# Theory of Mind Ability and Socioeconomic Status, a Study of Street-Connected Children and Adolescents in Ecuador

Graham Pluck [1,2]

1    Faculty of Psychology, Chulalongkorn University, Bangkok 10330, Thailand; graham.ch@chula.ac.th
2    Institute of Neurosciences, Universidad San Francisco de Quito, Quito 170901, Ecuador

**Abstract:** Family socioeconomic status (SES) is closely associated with cognitive ability in children and adolescents. However, most of the research has come from high-income countries. There is only limited research on 'street children', who represent an aspect of low-SES particularly associated with low- and middle-income counties. The current research in Quito, Ecuador, compared a group of street-connected youth with a not street-connected control group on two different measures of theory of mind ability and verbal comprehension. Initial analysis revealed that the street-connected sample scored significantly below the level of the control sample for verbal comprehension. For the main analysis, street-connected youth were matched to control participants for age, sex, and verbal comprehension scores. The street-connected sample was found to perform significantly below the control sample on both measures of theory of mind. Furthermore, worse performance appeared to be linked to severity of symptoms of post-traumatic stress disorder (PTSD) within the street-connected sample. In conclusion, the association of relatively poor verbal comprehension with street-connectedness is consistent with existing research from high-income countries on SES gradients and cognitive development. In contrast, theory of mind ability, a core aspect of social cognition, may be particularly linked to the street-connectedness form of low SES that exists in many low- and middle-income countries.

**Keywords:** street children; homeless youth; street connected youth; socioeconomic status; mentalizing; theory of mind; social cognition; verbal ability; language comprehension; post-traumatic stress disorder

## 1. Introduction

Research on the neurocognitive effects of variation in socioeconomic status (SES) has increased substantially in recent years. The research, mainly focused on children and adolescents, has led to a growing understanding that SES factors have a substantial impact on brain development [1–3] and cognitive functions [4–7], as well as aspects consequent on such development, such as adult attainment [8], and lifelong mental health [9]. However, most of the research has focused on socioeconomic disparity from a high-income country perspective (e.g., household income and education level examined as a gradient). This is understandable in that most peer-reviewed published research originates in such cultures. However, this has led to a dearth of research from low- and middle-income countries (LMICs). This is often considered to be a problem because the majority of published psychological and medical research is consequently focused on people who are actually only a small proportion of the human population [10,11], which may limit the generalizability of the observations.

In LMICs, one of the most obvious examples of socioeconomic deprivation are 'street children', more often referred to as street-connected youth in modern academic literature [12]. These are young people who spend long periods in the urban environment, unsupervised by adults, in the context of extreme poverty. The reasons for being in the street environment are varied. In some cases, youths are literally roofless/homeless and disengaged from parents, while in others, they are working (e.g., selling candies or shining

shoes), but have families and homes to return to. For most, the street environment is in some way or other a source of income, whether this is from honest trade, petty crime, or sex work, etc. The number of street-connected youth globally is unknown, but one well-controlled estimate of children aged 10–14 experiencing the most serious form of child homelessness, (i.e., literally roofless and disengaged from family) suggested that there may be about 10–15 million such individuals globally [13]. However, the actual figure for street-connected youth would be much larger than that if it were to include the multitude of young people in other street contexts, such as street-working children who live with their families.

Research on SES variables and cognitive ability in high-income countries tends to find that abilities are not uniformly associated with poverty. In particular, language, executive function, and declarative memory appear to be particularly linked to gradients in family SES variation [2,4,5], with lower family SES linked to lower task performance. The exact mechanisms underlying this association are not clear. One possibility stems from the fact that general cognitive ability is positively associated with measures of income (but not wealth, which is, to a large extent, inherited) [14]. Furthermore, as general cognitive ability, in the form of intelligence test scores, is highly heritable [15], SES, whether high or low, could be considered to be partly a consequence of normal biological variation in cognitive aptitude. On the other hand, children and infants from low SES backgrounds who move to higher SES families through adoption show very substantial increases in intelligence test scores [15], thus suggesting that variation in family SES background plays a causative role in cognitive ability development. Regardless of the mechanisms linking SES to child and adolescent cognitive ability, it is clear that there are strong associations, at least in the most developed countries [2,4–6,16,17].

Unfortunately, there is insufficient evidence about whether the same profile is observed in street-connected youth in LMICs. In a previous review of the extant literature, it was reported that there is an indication from intelligence-like tests that cognitive performance of street-connected youth in LMICs are below that of matched controls or normative data, and that the magnitude of difference from the comparison groups is more or less equivalent to that observed for homeless youths in the most developed countries [18]. However, the included studies were from diverse LMICs (Indonesia, South Africa, Colombia and Ethiopia) and there was substantial variation in results. Some street-connected youth samples showed a level that was only marginally lower than the comparison level (Indonesia), and some groups showed substantially lower scores (Colombia and Ethiopia). Since that review, two further studies have been published [7,19]; these again indicated a diversity of results. One reported on a group of Mexican street-connected youth who had a mean IQ score of 96, which is only moderately lower than the normative score of 100 [19]. Had the Mexican, rather than USA-based, norms been used, this score may have been above average. In contrast, a sample of Ecuadorian street-connected youth were found to perform tests of fluid intelligence and visuospatial ability substantially below a school-attending control sample [7].

However, in that study from Ecuador, it was also found that the street-connected sample performed relatively well on one particular measure, compared to their performance on other cognitive tests. That measure was the time-per-move ratio on the D-KEFS Tower Test [20], which appears to be a reliable test of executive functioning and is arguably the most valid metric of executive and prefrontal function from the numerous measures available for that test [21]. Similarly, a study in Bolivia compared homeless and similarly poor, but not homeless, youth on a range of cognitive tests. They found that the homeless youth performed as well as the not-homeless group on most measures, and actually significantly better on one particular test of executive function [22]. That was the Alternative Uses Test, an assessment which involves proposing alternative uses for common objects and is thought to measure creative, divergent thinking. The authors speculated that the street-connected youth in their sample may have developed better divergent thinking ability due to their experiences of homelessness.

While not specifically about street-connectedness, a recent study found that very poor South African children outperformed Australian children from substantially higher SES backgrounds on tests of executive functioning (inhibition and flexibility) [23]. The authors suggested that the large effects may be related to how South African children are expected to work within the home, respecting their roles, and their hierarchical place within the family, rather than following their own impulses, as is more common in the Australian context. Indeed, there is some evidence for work-related benefits to youth cognitive development. Brazilian youths working as vendors selling candies etc. in the streets may have better mathematical skills than age-matched school-attending children [24]. Again, the specific context is said to be the driving force, as vendors need to buy products wholesale and decide the prices that they want to sell at, as well as calculating the change to give for individual transactions. Similarly, precocious entry into the work environment in Nepal has been linked to better working memory performance of children, compared with those attending school [25], again, likely due to work experiences.

One has to be careful not to reify these specific cognitive benefits associated with street-connectedness and early experiences of work in LMICs, as in general, they exist within the context of generally worse cognitive functions, compared to more privileged youth. Nevertheless, their identification is important, as they represent ways in which different cultures and experiences impact on cognitive development. In particular, they help us to advance beyond the generalizations about SES and cognitive development gleaned from work in high-income countries, and recognize how diverse socioeconomic contexts influence neurocognitive development from a global perspective. This finer-grained analysis can have implications for how interventions, for example with at-risk individuals, are applied in different global contexts.

One area which may be of particular relevance to youth street-connectedness may be theory of mind ability, a core aspect of social cognition. Theory of mind, as a psychological construct, is the ability, also known as mentalizing, to determine the mental states of others and to empathize [26]. Although closely related to, and highly correlated with, executive functions, and likely relying on some of the same domain-general cognitive processes, theory of mind ability appears to be sustained by different neurobiological processes [27].

There are several reasons why street-connectedness might provide pressure for the development of the ability to read mental states in others. Street-connected youth in LMICs are frequently involved in the informal economy, such as begging, selling small items, or shining shoes. Theory of mind ability, as an individual difference factor, is known to influence achievement of sales personnel [28]. Those street-connected youth with better theory of mind ability may also perform better in the informal economy. Relatedly, this could potentially drive development of better theory of mind skill, similar to the ways that some cognitive functions of adolescents appear to be enhanced by entry to the workplace.

At the same time, street-connected youth are generally in very vulnerable situations and are at substantial risk of violence and exploitation [29,30]. The development of good theory of mind ability could be particularly useful for survival in hostile street environments. In fact, it is argued that theory of mind has evolved as a cognitive skill in humans precisely because individuals need to operate in social dominance hierarchies [31], where accurately recognizing the intentions of others is highly advantageous. It is possible that this ability might become well developed due to the challenges experienced and high-stake social interactions.

On the other hand, post-traumatic stress disorder (PTSD) is associated with poor theory of mind ability [32,33], and street-connected youth have high trauma exposure and may have increased risk of PTSD [19,34–36]. This would suggest the opposite association, that is, less well-developed theory of mind ability being linked to youth street-connectedness.

The aim of the current work was to examine whether social cognition in the form of theory of mind ability differs between children and adolescents with past street-connectedness and matched non-street-connected controls, in samples recruited in Quito, Ecuador (a middle-income country). The null hypothesis was that there would be no difference. A

secondary question addressed was whether PTSD symptoms would be associated with worse theory of mind task performance. The null hypothesis here was that the severity of PTSD symptoms would not be associated with worse theory of mind task performance. To preempt the results, I found no evidence that street-connectedness was associated with better theory of mind ability, and in fact, it appeared to be associated with worse task performance. Furthermore, the evidence suggests that PTSD symptoms may be linked to worse theory of mind ability in street-connected youth.

## 2. Results

### 2.1. Comparison of Street-Connected Youth with Their Control Sample for Verbal Comprehension

For the verbal comprehension scores in the Faux Pas Test, the street-connected sample (n = 37) scored a mean of 5.9 (SD = 1.7), which was lower than their control sample (n = 26), who scored a mean of 7.8 (SD = 0.4). This was a significant difference, Mann–Whitney $U = 859.00$, $p < 0.01$, median score difference = 2 points. As the comprehension difference would confound the interpretation of measures of theory of mind ability, in the next step, paired-sample analyses were implemented to control for differences in verbal comprehension.

### 2.2. Paired-Sample Analysis of 25 Street-Connected and 25 Control Matched Pairs

A paired-sample analysis was employed, in which individual street-connected youth were paired with control participants who had never been street workers or homeless. In addition to the primary data from street-connected children and adolescents, I also used the data set previously described for a study of SES and cognition [37]. In that study, the same measures of theory of mind were employed, that is, the 28-item Reading the Mind in the Eyes Test (RMET) and the 8-item Faux Pas Test. Both data sets were collected under supervision of the same doctoral level neuropsychologist, and around the city of Quito. Of that more recent SES sample, data on street-connectedness was collected from 49 participants. Seven of the forty-nine confirmed that they had been street workers. Thus, these were added to the street-connected sample from the original study, giving 37 + 7 = 44 street-connected participants. The other 42 participants from the SES study reported that they had never been street workers nor homeless. They were therefore added to the original control group, giving 26 + 42 = 70 participants without histories of street-connectedness (i.e., homelessness or having been street workers).

In order to compare performance on tests of theory of mind, while controlling for effects of verbal comprehension, I attempted to match each of the 44 street-connected youth in the combined sample with an individual with the same comprehension score from the combined control sample. The matching was strict; each member of a pair had to have the same comprehension score. When there was more than one potential match to a street-connected participant, matching was based on gender (male or female), and then on the closest age (which was calculated to two decimal places).

Of the 44 potential participants in the street-connected combined sample, only 25 could be matched to a control for exact comprehension score, and of those 25, all but two could also be matched for gender. Thus, the matching procedure produced 25 street-connected control matched pairs, with identical average comprehension scores (mean = 6.8, SD = 1.23, range = 4–8). Most of the matched controls (20/25, 80%) were found in the SES sample, and only 5/25 were from the original control group.

Twenty of the twenty-five (80%) street-connected pair members were male, compared to 18 of 25 (72%) in the control group members of the pairs. That small difference was not significant, $X^2 = 0.44$, $p = 0.51$, $V = 0.09$. Similarly, there was no significant difference for ethnic minority status between the groups; street-connected = 9/25 (36%) and control = 6/25 (24%), $X^2 = 0.86$, $p = 0.36$, $V = 0.13$. Nor was there a difference for age between the two groups; street connected mean = 14.1 (SD = 1.8) and control mean = 14.2 (SD = 1.8), $F(1,24) = 0.86$, $p = 0.36$, $\eta^2 = 0.03$.

The aim of this pairing procedure was to examine formerly street-connected youth and never-street-connected youth on measures of theory of mind, when there is no difference between the groups for verbal comprehension.

### 2.3. Psychometric Properties of the Assessments Used

As a first step, I determined the psychometric properties of the tests in the two groups. The psychometric properties of the 25 matched pairs are shown in Table 1.

**Table 1.** Psychometric properties of the of theory of mind tasks and verbal comprehension in the matched pairs of street-connected and control participants.

| Statistic | RMET | Faux Pas | Comprehension |
|---|---|---|---|
| | Street-connected (n = 25) | | |
| %CofV | 24 | 80 | 19 |
| $\alpha$ | 0.61 | 0.65 | 0.40 |
| Z Skew | 0.96 | 1.2 | 2.9 [1] |
| Z Kurtosis | 0.26 | 0.64 | 1.5 |
| | Control (n = 25) | | |
| %CofV | 21 | 46 | 18 |
| $\alpha$ | 0.64 | 0.75 | 0.43 |
| Z Skew | 1.4 | 0.54 | 2.9 [1] |
| Z Kurtosis | 0.87 | 1.1 | 1.5 |

[1] Values indicate significant deviation from a normal distribution.

For the 25 street-connected youth, all of the psychometric properties were within an acceptable range, with the exception that comprehension scores were significantly negatively skewed, caused by a ceiling effect, as 7 participants scored the maximum 8 points. This has acted to limit the variance in the data, and is a likely reason that the Cronbach's $\alpha$ value is quite low. As the matched control participants were matched on comprehension scores, the same issue is seen in the control sample psychometric values. When $\alpha$ values are low, this does not necessarily mean that the scale is not internally consistent; in this case, it is caused partly by the small number of items, 8, and low variance within the data. This issue has been previously addressed, and it has been shown that total scale scores can still be valid dependent variables, even when $\alpha$ values are suppressed due to homogeneity of variance, noting that $\alpha$ is actually the lower bound of internal consistency [38]. I therefore recalculated the $\alpha$ values using the formula of Pike and Hudson [39], which estimates the $\alpha$ value after accounting for the homogeneity of variance. In this case, if the data had been normally distributed, the estimated $\alpha$ value of comprehension scores for the street-connected youth would be 0.60, and for the control participants, 0.62. Although still rather low, they are just higher than the region suggesting 'unacceptable' internal consistency for research studies (i.e., $\alpha$ values < 0.60) [40].

### 2.4. Paired-Sample Comparison of Theory of Mind Performance

Mean scores on the two different measures of theory of mind ability are shown in Figure 1. The street-connected youth scored below the control participants on both assessments. These differences were significant for both RMET scores, $F_{(24,1)} = 9.04$, $p < 0.01$, $\eta = 0.27$, and for faux pas recognition scores, $F_{(24,1)} = 11.86$, $p < 0.01$, $\eta = 0.33$.

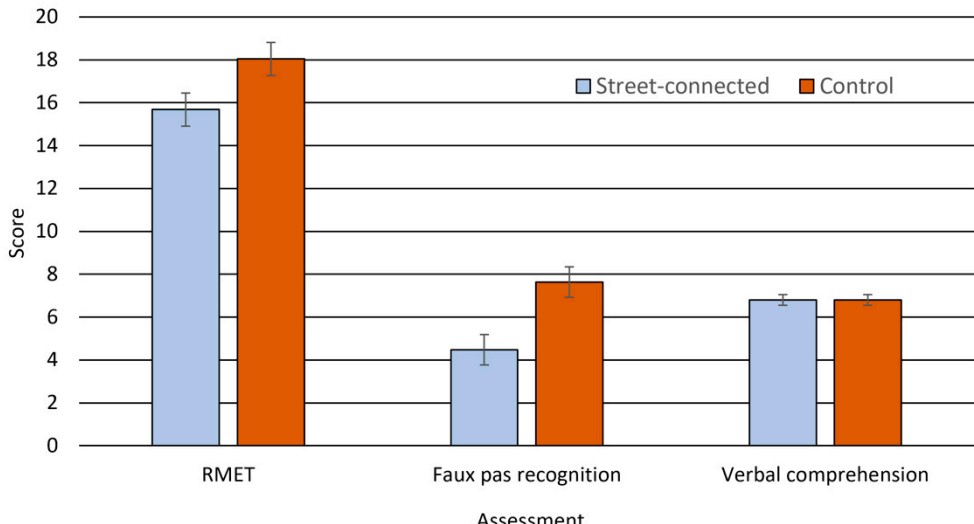

**Figure 1.** Mean (+SEM) for performance of theory of mind tasks and verbal comprehension by the 25 matched pairs of street-connected and control participants.

Although the two groups of 25 participants were matched for performance on verbal comprehension, 7 participants in each group scored a maximum of 8 points. It is therefore possible that comprehension was under-measured in one of the groups due to the ceiling effect, which would invalidate the matching procedure. I repeated the paired-sample analyses of theory of mind performance limiting the calculations to the 18 pairs in which comprehension scores were not at ceiling. This did not diminish the results. There was still a difference for RMET scores $F(17,1) = 4.28$, $p = 0.05$, $\eta^2 = 0.20$. This is now only borderline significant, however, that is likely due, in large part, to the reduced statistical power, as the effect size remains large. The significant difference for scores on the Faux Pas Test remained significant when limited to pairs with non-perfect comprehension performance, $F(17,1) = 18.90$, $p < 0.01$, $\eta^2 = 0.24$.

### 2.5. Correlations between Theory of Mind Performance and PTSD Severity Scores

All of the street-connected youth who were assessed with the UCLA-PTSD Reaction Index described past traumatic experiences. As Faux Pas recognition scores were correlated with age ($r = 0.27$) and RMET scores with both age ($r = 0.18$) and sex ($r = 0.39$, point-biserial correlation, 1 = male, 2 = female), these demographic factors were controlled for in partial correlations between theory of mind and PTSD severity scores. These revealed negative correlations. Higher PTSD severity was associated with significantly lower performance on the RMET ($r = -0.43$, $p = 0.04$). A similar pattern was seen with the Faux Pas recognition and PTSD severity score, but the correlation was not statistically significant ($r = -24$, $p = 0.18$).

### 3. Discussion

The street-connected youth reported here performed significantly below the level of a control sample on two different measures of theory of mind, one measuring perceptual-affective aspects (RMET) and the other measuring inferences about affective and epistemological mental states (Faux Pas Test) [41]. We therefore reject our null-hypothesis that street-connectedness is not related to theory of mind ability. The current research fills a gap in the research literature, providing novel evidence that theory of mind ability is lower than would be expected in street-connected youth.

In fact, the research was motivated by a question of whether street-connected youth, due to their experiences such as street working and homelessness, may have even developed better-than-control levels of performance. This was clearly not the case. The results therefore contribute to findings which, in general, suggest poor cognitive performance

related to street-connectedness of youth in LMICs [7,18]. In this sense, the results resemble those from high-income countries, which have identified various cognitive abilities including executive function and language skill as being linked to SES gradients [2,4–6]. However, it is unclear whether theory of mind ability is one of those abilities. Research with infants has provided conflicting results on whether SES in high-income countries is linked to development of theory of mind ability [6,42]. Research conducted on adolescents in Ecuador (a middle-income country), suggested that normal SES gradients are not related to theory of mind ability when language skill is controlled for [37].

If theory of mind ability is not related to SES gradients independently of language associations, then the observed results of significantly lower performance by the street-connected participants may be more due to the street-connectedness than to their low-SES. This interpretation appears likely, given the relatively high rates of trauma-related psychopathology in street-connected youths in LMICs [19,34–36], which is beyond that associated with living in low SES (but not street-connected) contexts. This is because post-traumatic stress is linked to poor theory of mind ability [32,33]. The current results are consistent with this interpretation, as negative correlations were observed between severity of PTSD symptoms and theory of mind task performance. The mechanism of this may involve the effects of stress and trauma on limbic system structures, particularly the amygdala [43], which is also closely linked to theory of mind ability [41]. Furthermore, the stress-related chronic elevation of cortisol levels is known to be linked to worse theory of mind task performance [37].

Care was taken in the current study to control for alternative interpretations. The data reported here was collected as part of a study on the same sample of street-connected youth previously reported [7,36]. However, at the time, journal reviewers raised concerns over the psychometric properties of the theory of mind measures, and the data was thus excluded from those published articles. I have since shown that the two theory of mind tests have acceptable reliability in the context in which we used them (i.e., Ecuadorian youths from low-SES backgrounds [37]. Additionally, I examined the reliability of the scales within the current samples, showing them again to be acceptable. This point is important, as it is becoming increasing clear that many cognitive tests have poor reliability, particularly when used in unusual contexts, which distorts findings when they are used as individual difference metrics [44,45], as I have used them here. The current results are unlikely to have been affected by this issue.

I also took additional steps to examine whether verbal comprehension could be the cause of the between-groups difference in theory of mind performance. This is also an important issue, as delayed or impaired language development is closely associated with socioeconomically excluded groups [16,46], and may be the true underlying cognitive factor driving apparent associations of SES with performance in cognitive tests that are ostensibly not dependent on language variation [5,6,37]. In the current context, this interpretation is unlikely, as the street-connected youths who performed significantly below controls were matched to have identical levels of measured verbal comprehension. A related question is whether variation in general cognitive ability could explain why the street-connected youth performed below the level of the comparison group. While this cannot be completely ruled out in the current study, the fact that the participants were matched for verbal comprehension again suggests that it is unlikely. This is because verbal skill is the cognitive factor which shows the most variation associated with SES [5,6,37], and is also the cognitive factor which best measures general cognitive ability [47]. It therefore seems likely that the groups in the current research were matched for the most pertinent cognitive factor.

There are implications of the current findings for care and interventions aimed at street-connected youth in LMICs. Firstly, street-connected youth, like homeless youth in high-income countries, may have relatively poor verbal ability compared to more privileged youth [17,48], which may have implications for engagement with services, recording of appointments, etc. Additionally, there may be some difficulties when it comes to social cognition, in particular reading the mental states of others (i.e., difficulties with theory

of mind). Many street-connected youths have had limited access to formal education. Although this is already a major focus of many agencies working with street-connected youth, the current results suggest that action to assist them to return to schooling will most likely improve verbal ability. Furthermore, literacy, in the form of reading fiction, is a known method to improve theory of mind ability [49]. As such, the current results provide empirical support for the aims of ongoing programs that emphasize the need to bring at-risk youths from the streets into formal education.

Some limitations of the current work should be addressed. The sample of street-connected youth described here will not be representative of all children and adolescents in street situations in LMICs. In particular, most of the individuals reported were already attending a charitable educational service, and were essentially formerly, not currently, street-connected. In this respect, they may differ in important ways from youth who remain in street situations and who may be not involved with educational and charitable programs aiming to assist them. An additional limitation is that the reported association between greater severity of PTSD symptoms and worse theory of mind task performance is based only on correlational analysis. It should thus be interpreted cautiously. I attempted to test the psychometric properties of the research assessments, particularly reliability, however, some doubts may remain about the validity of the tests in children and adolescents whose life experiences differ markedly from those that the tests were initially designed for; those in high-income countries. On the other hand, there is a dearth of research on this particular demographic. This is partly because of the general bias for published clinical and academic research to focus on the issues of WEIRD countries (i.e., Western, Educated, Industrialized, Rich, and Democratic) [11]. It is also partly due to the inherent difficulty in accessing young people who are not in contact with social, medical, and educational services, as is often the case for street-connected youth. The current research, in a small way, contributes to redressing this imbalance, by reporting on street-connected youths from a middle-income country in South America, and adding a diverse perspective to the ongoing research on socioeconomic influences on neurocognitive development.

## 4. Materials and Methods

### 4.1. Design

This research uses primary data on theory of mind test scores from a group of former street-connected 'street children' and a control group reported previously [7,36]. Sample sizes were chosen based on a calculation that at least 26 participants per group would be needed to detect, with a between-subject analysis, a significant difference at $p < 0.05$ (two-tailed), with a large effect size, at 80% power [50]. However, this data set is augmented with additional cases from a secondary data source, available at the PsychArchivs data depository, http://dx.doi.org/10.23668/psycharchives.2898, accessed on 1 September 2020. This data set corresponds to another study that I have conducted, on socioeconomic status and cognition [37]. Selected cases from that dataset were combined with the primary data to allow a paired-sample analysis.

### 4.2. Sample and Participants

4.2.1. Former Street-Connected Youth

The first sample, providing primary data, were 37 children and adolescents who were attending a charitable educational service which provides meals, education, and sporting activities to children and adolescents living in poverty in the city of Quito, Ecuador. All met a standard definition of 'street child' which proposes two basic situations, either of which must be fulfilled. In the first case, 'children of the street' is designated based on housing; these children are disengaged from their parents and living in the streets. The second situation, 'children in the street' is based on livelihood, and indicates children who work in the urban environment but live with their families [51]. The sample recruited here fit the second situation most clearly, albeit as former street children. This is because they were recruited from a service that requires daily attendance and cessation of street work,

and all were living with families or extended families at the time of recruitment. I have previously reported on trauma exposure in this sample: all had exposure to traumatic events, and 22/37 (59%) met criteria for current PTSD [36]. Meeting the definition of street child outlined above, being a speaker of Spanish and having a legal guardian who could provide consent to participate in the research were inclusion criteria; having a motor or perceptual disorder was the exclusion criterion.

### 4.2.2. Age-Matched School Attending Children and Adolescents

The second sample providing primary data was recruited as an age-matched control group for the street-connected youth described in Section 4.2.1. Twenty-six control participants were recruited from a state-run school in Quito. The same inclusion and exclusion criteria were employed. However, for this control sample, fitting the definition of street child, past or present, was an exclusion criterion. Participants in this control sample and the street-connected sample described in Section 4.2.1, all received a gift of stationery worth about USD 5 for participation. These samples have been described previously [7,36]. However, in those studies, only PTSD prevalence and cognitive data were reported. The theory of mind data reported here was not published at the time due to problems with scale psychometrics; issues which are addressed in the current report.

### 4.2.3. Children and Adolescents Recruited in a Study of Socioeconomic Status (SES)

A secondary data source was used to augment the data from the two primary samples described in Sections 4.2.1 and 4.2.2. This dataset is comprised of participants who were assessed as part of a study of SES and cognitive function [37]. However, in the second wave of data collection in that study, all participants were asked whether they had ever been street workers, or whether they had ever been homeless. The sample with street-connectedness data comprised 49 young people, all Spanish speakers. All participants in that study received a gift of stationery worth about US $10 after participating. However, it should be noted that many of these cases are not reanalyzed in the current research, but they were included in the overall pool of potential participants when forming a paired-sample, as described in Section 2.2.

### 4.3. Assessments

The participants in all three samples completed two different tests of theory of mind, available in Spanish from the creator of the tests at https://www.autismresearchcentre.com, accessed on 19 December 2011.

The first of these was the children's version of the Reading the Mind in the Eyes Test (RMET) [52]. There are 28 trials, and each involves examining a greyscale photograph of the eye region of an individual presented in the center of an A4/US letter size page. In each corner of the page is a word or phrase describing the possible mental states of the person in the image. The participant attempts to choose the one correct option on each trial. This is considered to be a perceptual measure of affective theory of mind [41]. This test is widely used to measure theory of mind in research contexts. Nevertheless, the RMET tends to have poor internal consistency, as measured with Cronbach's alpha (range of $\alpha$ between 0.37 and 0.63), although it does appear to be unidimensional of a factor defined as 'perspective taking' [53]. We have previously shown in an Ecuadorian youth sample that the internal consistency is perhaps slightly better than in other studies, though still rather low at $\alpha = 0.61$, but with a slightly better retest reliability of $r = 0.70$ [37].

The other theory of mind test employed was the Faux Pas Test [54]. The eight-scenario version used here has been previously described [37]. Each scenario is read aloud to the participant, and it is also displayed in large type on an A4/US letter size page placed in front on them. The text uses simple vocabulary, and the scenarios vary in length from 32 to 106 words. The scenarios describe social situations with dialogue between 2 or 3 people, such as two children discussing a party that they are going to attend. In four of the scenarios, one of the characters unwittingly says something that upsets one of the

other characters (i.e., they make a faux pas). In the other four scenarios, there are no faux pas. The participant is asked of each scenario whether somebody said something that they should not have said; this question probes the ability to recognize affective mental states. If they correctly identify a scenario containing a faux pas, they are asked two additional questions. One is to describe what was said that should not have been, and the other is to probe whether the participant understood the epistemic mental states by identifying the false belief held by a character that led to the faux pas. As there are four faux pas containing scenarios, with three faux pas-related questions in each, the potential score range for faux pas recognition is 0 to 12 points.

In each of the eight scenarios, there is a single control question that probes comprehension of the story, which does not require mentalizing to answer, such as where the characters were located during the conversation. The comprehension score therefore potentially ranges from 0 to 8 points. It has previously been shown in an Ecuadorian youth sample that the faux pas recognition score has good internal consistency ($\alpha = 0.82$) and retest reliability ($r = 0.79$) [37]. However, for the comprehension scale, meaningful estimates of internal consistency and retest reliability could not be produced, due to a ceiling effect, with the majority of participants making less than 2 errors (the modal score was 8/8 correct).

The sample of former street-connected youth (described in Section 4.2.1) were all assessed for PTSD symptoms with the UCLA PTSD Reaction Index for DSM IV-Adolescent Version Spanish [55]. This is a semi-structured interview comprising 49 items. I previously reported the prevalence of PTSD data in that sample [36]. However, the assessment also allows the derivation of a severity index score, which I did not include in the previous study. In the development sample for the schedule, the severity index score was found to have excellent internal consistency (Cronbach's $\alpha = 0.90$) and convergent validity by its correlation with an established scale of post-traumatic stress ($r = 0.75$) [55].

### 4.4. Procedure

In accordance with the ethics committee-approved protocols, all children and adolescents provided written informed assent to participate. In addition, written informed consent to recruit the individuals was taken from a parent or guardian. The research was conducted in accordance with local laws and research guidance provided by the American Psychological Association and the Declaration of Helsinki. All participants were assessed on a larger battery of tests that has been reported previously, including measures of executive function and intelligence [7,36]. In each assessment, the two theory of mind tests reported here were within the last three tests to be administered within a session, which took 50 to 60 min to complete. For the sample of formerly street-connected youth only (described in Section 4.2.1), each participant was interviewed with the UCLA PTSD Reaction Index. Each data collection session was conducted in a quiet, private room, by a research assistant supervised by a doctoral-level neuropsychologist. On completion of the tests, all participants were debriefed and were able to ask questions. They were thanked and given gifts for participation.

### 4.5. Statistical Analysis

Raw scores were used for all analyses. To aid the interpretation of the data distributions and psychometric properties, we calculated coefficients of variance expressed as percentages (%CofV), Cronbach's $\alpha$ values, and z scores of the skew and kurtosis. There are no cut-off scores for %CofV when calculated from the individual mean scores from a group on a particular test. However, relatively lower scores compared to a different test, or a different sample, show that between-person variance is lower, in turn suggesting that the data set will perform relatively less well as an individual difference metric [44,45]. Regarding interpretation of Cronbach's $\alpha$, although standards are more stringent in scale development, for research purposes, $\alpha$ values $\geq 0.60$ can be acceptable [40]. Regarding skew and kurtosis, if both z scores are less than 1.96, equivalent to a > 0.05 significance

threshold, the data can be considered normally distributed [56]. In such instances, data were analyzed with ANOVA with effect sizes given as $\eta^2$, else, non-parametric methods were used, with the effect size given as absolute median difference in scores. All analyses employed a significance threshold of 0.05. Hypotheses in which directional effects were not specified were analyzed with two-tailed tests, or one-tailed when the hypothesis was directional. Cramer's *V* is used for effect sizes of categorical data analyses. All inferential tests were conducted with SPSS version 23.0.

**Funding:** This research received no external funding.

**Institutional Review Board Statement:** The study was conducted according to the guidelines of the Declaration of Helsinki, and approved by the Institutional Review Board of Universidad San Francisco de Quito (protocol code 2011-52) on 21 September 2012.

**Informed Consent Statement:** Informed consent was obtained from all participants involved in the study.

**Data Availability Statement:** The primary data supporting this research are available from the author on request. The secondary data reported are available at PsychArchives, http://dx.doi.org/10.23668/psycharchives.2898 (accessed on 20 April 2021).

**Acknowledgments:** I would like to thank the research assistants involved in this study: Victoria Andrade-Guimaraes, Daniel Banda-Cruz, Sofia Ricaurte-Diaz, Doenya Amraoui, Isabella Fornell-Villalobos, and Christine Bock.

**Conflicts of Interest:** The author declares no conflict of interest.

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
