# Peer review of "Theory of Mind Ability and Socioeconomic Status, a Study of Street-Connected Children and Adolescents in Ecuador"

_psych, doi:10.3390/psych3020008_

Round 1

Reviewer 1 Report

The author compared the performance on tests assessing the theory of mind of a group of street-connected children with a control group. The results showed that street-connected children performed worse on this type of test.

General comment: While I acknowledge the author for having tackled an important subject and for the statistical analyses carried out with great precision, the work does not convince me with respect to either novelty or methodology. Furthermore, the presentation of the introduction as well as the discussion of the results have obvious shortcomings. In addition, the work is dripping with redundant and unnecessary information, both in terms of statistics and the description of participants who are not used. The work needs to be cleaned up and perhaps all the technical information should be included in an appendix to the work.

Other comments:

  1. Lines 7-10 the English sentence construction is not accurate making the introduction of the abstract difficult to read and understand;
  2. Line 10 the author talks about the work being conducted in the first person, a procedure strongly discouraged by the scientific community;
  3. Line 11 the concept of street-connectedness is introduced but it is not clear what it refers to;
  4. Line 13 what does the matched control group refer to, controlled for what?
  5. Overall, the abstract is unclear. In short, the abstract needs to be rewritten clarifying the theoretical background, objectives, constructs used, sample, results and conclusions. English, specifically sentence construction, needs to be revised.
  6. Line 31, unintelligible English (i.e. as aspects consequent such development such as adult attainment).
  7. Line 34-36 why is there a criticism that most of the research comes from high-income countries? What would be the point?
  8. Line 54-56 the author mentions his/her own research, which I repeat should not be done except in the third person, but there is no reference to it;
  9. Line 62 the author mentions two other studies but the bibliographical references are missing. This is repeated throughout the text at some points. Care should be taken to ensure that all works referred to have a reference and to avoid continual reference to one's own work;
  10. Line 75 suddenly the author introduces "the Alternative Uses Test", but it is not clear why or in what context.
  11. Line 110-115 it is not clear how the author arrives to formulate the hypotheses, only when he/she anticipates the results the reader can infer that the author expected to find a superior theory of mind in street-connected children. But why does the author expect these results?
  12. Has the research conducted been approved by which ethics committee? Has the author obtained legal consent to disseminate the results?
  13. What does "identified as male" mean? Is it not enough to say that N children were male?
  14. Sub-section 2.2.3 the sample is characterized by participants of a different age with respect to the other two samples and it is not clear how and why it is used since the other participants have been tested in another time and context. Unfortunately it turns out to be too heterogeneous a sample to be able to deduce anything meaningful and reliable. It is only later that it becomes clear that only 25 participants were drawn to match the first group. But then why present the whole sample? What information does the reader get.
  15. Statistical analyses have been accurately carried out. However, I find it difficult to understand why analyses have been carried out on scales that have shown low internal consistency. What did the author hope to achieve? It is obvious that if the results on the scales are unreliable, any statistically significant differences that might emerge are not intelligible.... why not just present the data from the second analysis?
  16. In general, all the statistics could be relegated to an appendix because it is really redundant, sometimes unnecessary and confuses the reader.
  17. As for the discussions, it is not clear how the results represent a novelty compared to previous studies. Moreover, the question concerning the tests used to evaluate the theory of mind does not seem to be resolved at all. In fact, the author himself clearly showed that in the first analyses the data were not reliable.

Author Response

File attached

Reviewer 2 Report

Summary: The author explores whether street-connectedness is related to theory-of-mind performance. It’s suggested that street children might have elevated theory-of-mind ability, since many engage in trades which would benefit from an ability to read customers. It’s found, however, that the street children perform worse on theory-of-mind tasks. The author relates these results to literature on the effect of trauma, commonly experienced by street children, on theory-of-mind performance.   

Comments: This is a well written paper on an interesting topic. Samples are detailed well, as was the protocol, test procedure, and materials. The analysis was well thought out. And potential statistical concerns were ably addressed. The conclusion is appropriate.

It would help interpretation if the author could report the age-relevant means from the test battery standardization samples. Otherwise, we are left comparing two convenience samples -- and one or the other (or both) may be unrepresentative.   

The writing is generally lucid, but there is still some awkwardness in phrasing. I will give some examples below and ask for language proofreading / editing.

Examples:

“The reasons for being in the street environment are varied, and include being literally roofless/homeless and disengaged from parents, as well as young people who are working, for example selling candies or shing shoes, but who have families and home to return to.”

The above lacks parallel construction but could be rewritten as e.g., :

“The reasons for being in the street environment are varied. In some cases, youths are literally roofless/homeless and disengaged from parents, while, in others, they are working (e.g., selling candies or shining shoes) but have families and homes to return to.”

“petty crime, sex work etc”  <-- comma after sex work

“Some street-connected youth samples showing a level that was only marginally lower than the comparison level (Indonesia), and some groups showing substantially lower scores (Colombia 62 and Ethiopia).” <-- replace “showing” with “showed”

“. Since that review two further studies have been published, these again indicated diversity of results.” <-- Try: “Since that review, two further studies have been published, which again indicated a diversity of results.” [Missing article]

“One reported on a group of Mexican street-connected youth had a mean IQ score 64 of 96 which is only moderately lower than the normative score of 100 [12], and may have even 65 surpassed it if Mexican rather than USA-based norms had been used” < --Try: “One reported on a group of Mexican street-connected youth who had a mean IQ score of 96, which is only moderately lower than the normative score of 100 [12]. Had the Mexican, rather than USA-based norms been used, this score may have been above average.”

“Similarly, precious entry into the work environment”< -- “precious entry”? What does this mean?

“To address those issues, in the other phase of analysis we used additional cases form a secondary data source”  <-- “from” a secondary; you also confuse “form” and “from” elsewhere.

“The first situation, ‘children of the street’ is based on housing, and indicates children, disengaged from parents, for whom the street situation is their habitual abode (i.e., homelessness). <-- Try e.g.,: In the first case, “children of the street” is designated based on housing; these children are disengaged from their parents and living in the streets.

“, motor or perceptual disorders were exclusion criteria.” <-- This is a run on sentence. Try e.g.,: “; having a motor or perceptual disorder was the exclusion criterion.”

Note, don't feel obliged to use my rephrasing; but rewrite these passages and do some more language-related editing. 

Author Response

File attached

Reviewer 3 Report

The manuscript is interesting and the topic of the study is important. In my opinion, only relatively little improvement is needed for the article to be accepted.

There should be a spell-check. For instance, I noticed that in line 404, the word "may" should perhaps be "many cognitive tests have poor reliability".

For some reason, genetic background is not discussed at all, despite intelligence-related traits being highly heritable in developing countries. It would be very interesting and important for the reader to know, what is the heritability of different cognitive or theory of mind-related skills in developing or middle income countries, especially among street connected children or among children with a low SES background. It is commonly assumed that in very developed countries the heritability of a trait actually goes up as the school system evens out the environmental differences between individuals. The manuscript does not even mention the possibility that poor cognitive skills could be a driver or partial reason why some children are more likely to end up being street-connected. However, it is likely, for instance, that if a child is not able to study in a regular school, they will more likely become street-connected. Perhaps some additional variable or variables should have been collected to address this issue. At least it should be mentioned.

The literature review should provide the reader with some examples of the associations between cognitive skills and theory of mind - is this association strong or weak in general? The author did not explicitly state if the results could be predicted from a lower general cognitive ability.

The study mentions several times that PTSD lowers cognitive skills and/or theory of mind and that it is common among street-connected children. Why was this or other relevant psychological traits not measured from the children of the original samples or from the SES-group? This issue should be addressed. It is highly relevant whether or not the lower scores can be attributed to PTSD or mental-health (or even personality) -related variables.

The presentation of the results should be improved so that a "busy reader" would see the main results with the first glance. Perhaps a figure or a different format table would help.

In general I believe the study topic is very important and after some revision this manuscript should be published.

Author Response

File attached

Round 2

Reviewer 1 Report

The author has made the manuscript clearer and sufficiently addressed all my previous criticisms. 

Author Response

Thank you

Reviewer 3 Report

I think the manuscript has improved and is suitable for publication.

Author Response

The manuscript has been reviewed for English language use and style